# Relationship between weekends catch-up sleep and risk of aging

Nan Yao[1]☉, Liuyi Shen[2]☉, Linrui Qi[3], Wenqiang Li[1], Chenan Liu[4], Fuzhou Han[1], Ning Duan[1], Guoyong Yu[5]*, Jun Qu (ID)[1]*

1 Department of General Surgery, Aerospace Center Hospital, Beijing, China, 2 Department of Orthodontics, School and Hospital of Stomatology, Hebei Medical University, Shijiazhuang, China, 3 Department of Neurology, The First Affiliated Hospital of Jinan University, Guangzhou, China, 4 Department of Gastrointestinal Surgery, Department of Clinical Nutrition, Beijing Shijitan Hospital, Capital Medical University, Beijing, China, 5 Department of Nephrology, Beijing University of Chinese Medicine Affiliated Dongzhimen Hospital, Beijing, China

☉ These authors contributed equally to this work.
* qujunchief@163.com (JQ); 18901133535@163.com (GY)

## Abstract

### Background

Sleep has been proven to be associated with various chronic diseases and aging. However, many individuals fail to achieve recommended sleep durations on weekdays and opt for compensatory sleep during weekends. This study aims to investigate the relationship between weekend catch-up sleep (CUS) and aging.

### Methods

All participants were sourced from NHANES 2017–2018. Using the sleep questionnaire, we obtained participants' sleep timings and durations on weekdays and weekends. Weekend CUS was identified as an extension in average weekend sleep duration. Biological age is a biomarker for evaluating biological aging, and its difference from actual age is used to determine aging. Weighted logistic regression analysis was employed to explore the relationship between CUS and aging.

### Results

A total of 4,713 participants were included in this study, with an average age of 47.54±16.94 years. 50.6% of individuals experienced CUS. Compared to individuals without CUS, participants with CUS had a 20% lower risk of aging (OR = 0.80, 95% CI: 0.63−1). Specifically, participants who engaged in CUS for 0−1 hour showed a 23% lower risk of aging (OR = 0.77, 95% CI: 0.61–0.96), and those with CUS for 1−2 hours had a 20% lower risk of aging (OR = 0.80, 95% CI: 0.65–0.98). Stratifying by bedtime, the relationship between CUS and reduced aging risk is only observed in individuals who usually go to sleep before midnight and have CUS less than 2 hours.

**Data availability statement:** "All data can be applied for through the official website of NHANES (https://wwwn.cdc.gov/nchs/nhanes/Default.aspx)".

**Funding:** The author(s) received no specific funding for this work.

**Competing interests:** The authors have declared that no competing interests exist.

**Abbreviations:** CRP, C-reactive protein; CUS, catch-up sleep; NCHS, National Center for Health Statistics; NHANES, National Health and Nutrition Examination Survey; OR, odds ratio; PIR, ratio of family income to poverty; RCS, restricted cubic spline.

## Conclusion

The 0–2 hour CUS is associated with a reduced risk of aging, and this relationship is more significant in participants who go to bed early and have healthy sleep patterns.

---

## Introduction

Sleep is closely related to health, and maintaining a regular sleep pattern, stable sleep duration, and high-quality sleep is crucial for overall well-being and disease resistance, which includes the impact on cardiovascular diseases, cognition, metabolism, and various other conditions such as tumors [1,2]. Numerous studies have recommended optimal sleep duration and patterns. A study by Li et al. from the UK Biobank, based on the non-linear association between sleep duration and genetic, cognitive, brain structure, and mental health factors, found that the optimal sleep duration was approximately 7 hours [3]. Furthermore, another study, analyzing two independent cohorts, demonstrated an association between a healthy sleep pattern including timing, duration, snoring, and excessive daytime sleepiness and a lower risk of heart failure, independent of traditional risk factors [4]. Some prospective studies also indicated a close association between excessively long or short sleep duration and an increased risk of cancer [5,6].

However, in recent years, the rapid development of society, coupled with the demands of work and irregular lifestyles, has led to an increase in staying up late and a reduction in sleep duration [7,8]. As a result, many people have resorted to opt for Catch-Up Sleep (CUS) during the weekends to compensate for the lack of sleep on weekdays. This concept is similar to the "weekend warrior" idea proposed by Dos et al., where many adults, due to factors such as work or overtime, are unable to meet the daily requirements for physical activity [9]. Instead, they opt to engage in equivalent intensity physical activity for one or two days each week to mitigate the harms associated with prolonged sitting and low physical activity [10]. This approach aims to achieve benefits and lower mortality rates similar to those who engage in regular physical exercise. Recent studies have defined various criteria for measuring catch-up sleep (CUS). A study based on the National Health and Nutrition Examination Survey (NHANES) defined CUS by the difference in sleep duration between workdays and weekends, categorizing it into no CUS, mild CUS (0-1h), moderate CUS (1-2h), and long CUS (>2h) [11,12]. CUS appears to be associated with certain diseases; for instance, Liu et al. demonstrated that individuals with 1–2 hours of CUS had a lower risk of depression compared to those who did not have CUS [12]. Chen et al. showed that compared with those without CUS, 2–3 hours of CUS was closely associated with a lower prevalence of chronic kidney disease [13]. However, its relationship with other diseases or aging remains unclear.

Aging is a collective term for the decline in physiological functions during adulthood, characterized by features such as increased susceptibility to diseases, genomic instability, epigenetic changes, and chronic inflammation [14,15]. Despite clear molecular-level characteristics of aging, it is challenging to precisely define

whether an individual has truly undergone aging at the population level. In fact, recent research encourages evaluating whether aging has occurred at the population level by examining the difference between physiological age and chronological age [14,16]. Coincidentally, Liu et al. developed a widely validated and used phenotypic age calculation formula based on NHANES data [17]. Up to now, although no studies have explored the relationship between the increasingly common weekend catch-up sleep (CUS) habit and aging, research has investigated the association between sleep duration patterns and aging. The results showed that compared with participants whose sleep duration was in a normal and stable trajectory, trajectories of increased sleep duration and short-term stable sleep duration were associated with a lower likelihood of successful aging. Worse sleep patterns were also closely related to cognitive decline, yet the relationship between catch-up sleep and aging remains unaddressed [18,19]. To fill this gap, this study aims to investigate the association between CUS and aging using the NHANES database, and further explore the anti-aging effects of weekend CUS among individuals with different sleep times, sleep durations, and sleep disorders.

## Methods

### Study population

As reported in our previous research, NHANES is a large-scale cross-sectional study conducted by the National Center for Health Statistics (NCHS). It aims to investigate the health and nutritional status of adults and children in the United States [20].

In this study, data from the year 2017–2018 were utilized, as during this survey cycle, the majority of participants reported their average sleep duration on workdays and weekends, as well as issues like snoring and sleep disorders. Among these patients, 4,771 individuals had information on variables used to calculate phenotypic age. Out of these, 11 participants lacked information on educational level, 4 lacked information on marital status, 36 were missing sleep duration data, and 7 were missing sleep timing information. Ultimately, 4,713 participants were included in this study.

### Exposure assessment

The main exposure factor is CUS, derived from participants' responses to the NHANES questionnaire regarding average sleep duration on weekdays and weekends. An increase in average weekend sleep duration is considered as CUS. Additionally, the study distinguishes the extent of CUS based on the difference values, we classified CUS duration as≤0 hours (i.e., no CUS), 0<CUS duration≤1 hour, 1 hour<CUS duration≤2 hour and >2 hour [8,9]. This information will be utilized to assess the relationship between weekend CUS and other factors such as phenotypic age and anti-aging effects [7].

### Outcome

This study's primary outcome is aging. Initially, phenotypic age is calculated based on previous research. Factors within the equation include albumin, alkaline phosphatase, C-reactive protein, total cholesterol, blood creatinine, glycated hemoglobin, systolic blood pressure, blood urea nitrogen, uric acid, white blood cell count, lymphocyte percentage, mean cell volume, and red blood cell distribution width [21]. The specific calculation methods are detailed in Supplementary Method 1 [22]. Subsequently, the calculated phenotypic age is subtracted from the actual age. If the difference is greater than 0, it is categorized as aging; otherwise, it is classified as non-aging.

### Covariates

Sleep-related content has been adapted from the Munich ChronoType Questionnaire, covering sleep habits, sleep disorders, sleep duration, and wake-up times [23]. Participants are typically asked, "What time do you usually fall asleep/wake up on weekdays or weekends?" Based on the sleep time and using midnight (00:00) as the reference point, participants are categorized as either early sleepers or late sleepers. Sleep duration is categorized based on previous research.

Participants are divided into three groups: < 7 hours, 7–8 hours, and >8 hours, and are respectively considered to have short, normal, and long sleep durations [24]. Sleep disorders are recognized by healthcare professionals, and participants are asked, "Have you ever told a doctor or other health professional that you have trouble sleeping?". Covariates in the study include age, gender, marital status (Married, Separated, Never married), BMI (considered obese if greater than or equal to 30 kg/m2), Ratio of family income to poverty (PIR, < 1.30, 1.30–3.49, ≥ 3.50) [25], education level, smoking, alcohol consumption, and physical activity. Data on physical activity are derived from the Global Physical Activity Questionnaire, and all adults provided interview data on physical activity, including vigorous activities, moderate-intensity exercise, and more [26].

### Statistical analyses

All statistical analyses were conducted using R 4.2.3. A two-sided P-value of <0.05 was considered statistically significant.

Weighted analysis was employed in the analyses, implemented using the "survey" package in R [27]. Continuous variables were presented as mean ± standard deviation (SD) or median (P25, P75), and between-group differences were assessed using t-tests or Mann-Whitney U tests depending on the data distribution (normal or non-normal). Categorical variables were expressed as N (%), and between-group differences were compared using the chi-square test. Restricted cubic spline (RCS) plots were used to illustrate the nonlinear relationship between aging risk and sleep duration or weekend CUS duration. Weighted logistic regression analysis was employed to investigate the relationship between aging risk and weekend CUS, with results described in terms of odds ratios (OR) and 95% confidence intervals (CI). We conducted three models in the weighted logistic regression analysis. Model 1 represents the crude model, Model 2 adjusted for age, sex, marital status, PIR (Ratio of family income to poverty), educational level, and obesity. Model 3 adjusted for age and sex, marital status, PIR, educational level, obesity, smoke, alcohol use, sleep trouble, PA, sleep duration in work day (Supplementary Method 2). Considering that the timing and duration of sleep may also be related to aging, we further investigated the interaction between the timing and duration of sleep in the context of CUS and aging. CUS duration could act as a mediator between weekday sleep patterns (e.g., total weekday sleep, bedtime) and aging risk. Therefore, mediational analysis was used to clarify the role of CUS in these relationships.

Interaction analysis was used to explore whether CUS is a moderator of the relationship between weekday sleep duration or weekday bedtime and aging. Additionally, we conducted subgroup analyses and sensitivity analyses, including exploring the effects of CUS in participants with and without sleep disorders, as well as in those who were retired. Sensitivity analysis involved excluding patients taking sleep medications and antipsychotic medications (psychotherapeutic agents, stimulants, glucocorticoid, benzodiazepines, non – benzodiazepine hypnotics, miscellaneous anxiolytics, sedatives and hypnotics, benzodiazepine anticonvulsants, phenothiazine antipsychotics,ssri antidepressants and hypnotics, antiemetics), participants with night shifts or participation without a work schedule provided, participants with CUS less than −30 minutes.

## Results

### Baseline characteristics

A total of 4,713 participants (weighted: 102,908,622) were included in this study, with an average age of 47.54 ± 16.94 years. Among them, 2,267 were men, and 2,446 were women. 50.6% of individuals experienced CUS. Compared to those who did not catch up on sleep, participants with CUS were younger and more likely to have higher income or consume alcohol (Table 1). Regarding sleep duration, age showed a U-shaped correlation with weekday sleep duration and a negative relationship with weekend sleep duration (S1 Fig in S1 File). As for CUS duration, the proportion of individuals engaging in CUS increased initially with age, peaking in the 30–40 age group, and then declined. However, the proportion of individuals with CUS greater than 2 hours increased with age, corresponding to a decrease in this category (Fig 1).

**Table 1. Baseline characteristics.**

| | Overall (Weighted) | Without CUS (Weighted) | With CUS (Weighted) | p |
|---|---|---|---|---|
| N (Weighted) | 4713 | 2613 (49.4) | 2100 (50.6) | |
| Age (mean±sd)# | 47.54 (16.94) | 50.92 (19.23) | 43.36 (14.54) | <0.001* |
| Phenoage (mean±sd)# | 46.19 (17.99) | 51.82 (18.12) | 41.57 (15.36) | <0.001* |
| Sex (%) | | | | 0.023* |
| Men | 2267 (49.7) | 1290 (52.6) | 977 (46.9) | |
| Women | 2446 (50.3) | 1323 (47.4) | 1123 (53.1) | |
| BMI (mean±sd)# | 29.58 (6.83) | 29.21 (6.38) | 29.94 (7.23) | 0.159 |
| Obesity (%) | | | | 0.117 |
| No | 2761 (58.1) | 1585 (60.9) | 1176 (55.4) | |
| Yes | 1952 (41.9) | 1028 (39.1) | 924 (44.6) | |
| Marital status (%) | | | | 0.023* |
| Married | 2809 (64.4) | 1504 (62.4) | 1305 (66.4) | |
| Separated | 1062 (16.5) | 683 (19.8) | 379 (13.3) | |
| Never married | 842 (19.1) | 426 (17.8) | 416 (20.3) | |
| PIR (%) | | | | 0.030* |
| <1.30 | 1149 (14.8) | 696 (17.9) | 453 (11.8) | |
| 1.30-3.49 | 1703 (30.3) | 962 (30.6) | 741 (30.0) | |
| ≥3.50 | 1861 (54.9) | 955 (51.4) | 906 (58.2) | |
| Educational level (%) | | | | 0.127 |
| Below college | 3593 (75.7) | 1974 (74.3) | 1619 (77.2) | |
| College or above | 1114 (24.3) | 634 (25.7) | 480 (22.8) | |
| Alcohol (%) | 3053 (75.1) | 1597 (71.8) | 644 (78.3) | 0.014* |
| Smoke (%) | 1132 (25.1) | 706 (26.2) | 426 (23.9) | 0.531 |
| PA (%) | 3509 (81.4) | 1877 (80.9) | 1632 (82.0) | 0.511 |
| Sleep trouble (%) | 1347 (28.6) | 806 (28.3) | 541 (28.9) | 0.831 |
| CUS time (median [IQR])§ | 0.50 [0.00, 1.00] | 0.00 [−0.50, 0.00] | 1.00 [1.00, 2.00] | <0.001* |
| Sleep duration in workdays (median [IQR])§ | 7.50 [7.00, 8.00] | 8.00 [7.00, 8.00] | 7.50 [7.00, 8.00] | <0.001* |
| Sleep duration in weekends (median [IQR])§ | 8.00 [7.50, 9.00] | 7.50 [7.00, 8.00] | 9.00 [8.00, 9.50] | <0.001* |

CUS, catch-up sleep, BMI, body mass index, PIR, poverty income ratio, PA, physical activity.

*Indicates statistical significance.

# Indicates that the inter-group difference was tested using the t-test.

§ Indicates that the inter-group difference was tested using the Mann-Whitney U test.

## The relationship between CUS and aging

RCS analysis reveals a U-shaped nonlinear relationship between the duration of CUS and the risk of aging, indicating that moderate CUS duration may be more beneficial for reducing aging risk (S2 Fig in S1 File). The subsequent logistic regression results (Table 2) showed that the OR values between CUS and aging risk were 0.79 (95% CI: 0.64–0.97), 0.77 (95% CI: 0.62–0.99), 0.80 (95% CI: 0.63–1) in Model 1, Model 2, and Model 3, respectively. In detail, when subdividing the duration of CUS and comparing it with participants who did not engage in CUS, those with CUS durations of 0–1 hour (OR=0.77, 95% CI: 0.61–0.96) and 1–2 hours (OR=0.80, 95% CI: 0.65–0.98) had a 23% and 20% reduced risk of aging, respectively. However, participants with excessively prolonged CUS (>2 hours) did not exhibit a significant reduction in the risk of aging (OR=1.06, 95% CI: 0.66–1.70), indicating that the protective effect of CUS on aging may disappear when the duration exceeds 2 hours.

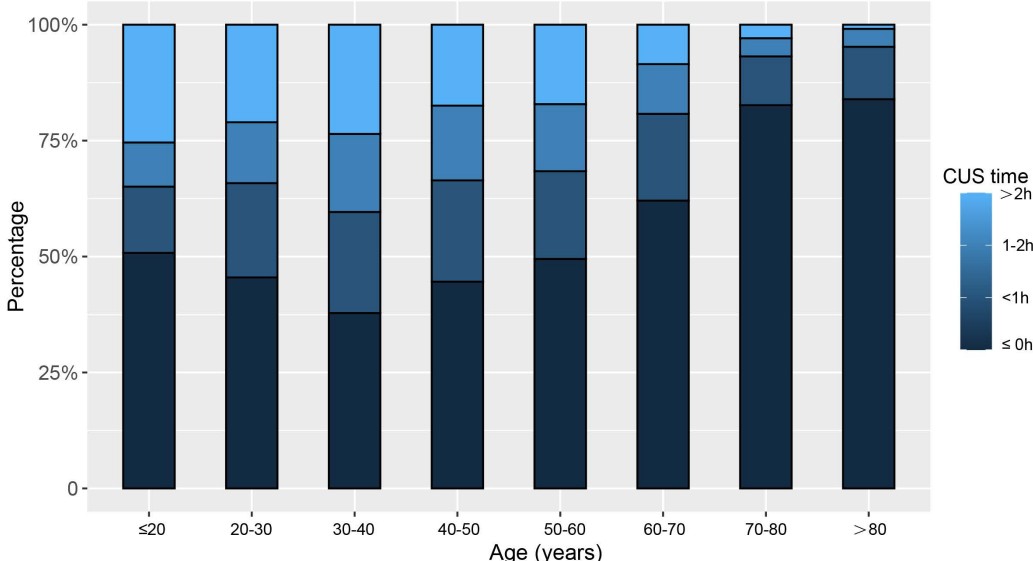

**Fig 1. Percentages of CUS between different ages.**

**Table 2. The relationship between weekend CUS and aging risk.**

| | Model 1 | | Model 2 | | Model 3 | |
|---|---|---|---|---|---|---|
| **Weekends CUS** | **OR (95% CI)** | **P** | **OR (95% CI)** | **P** | **OR (95% CI)** | **P** |
| No | Ref. | | Ref. | | Ref. | |
| Yes | **0.79 (0.64, 0.97)** | **0.026*** | **0.77 (0.62, 0.97)** | **0.032*** | 0.80 (0.63, 1) | 0.051 |
| Weekends CUS duration (h) | | | | | | |
| ≤0 | Ref. | | Ref. | | Ref. | |
| 0-1* | **0.71 (0.58, 0.88)** | **0.004*** | **0.76 (0.60, 0.96)** | **0.016*** | **0.77 (0.61, 0.96)** | **0.031*** |
| 1-2* | **0.72 (0.61, 0.85)** | **0.001*** | **0.74 (0.60, 0.90)** | **0.004*** | **0.80 (0.65, 0.98)** | **0.045*** |
| >2 | 1 (0.67, 1.48) | 0.990 | 0.99 (0.63, 1.56) | 0.946 | 1.06 (0.66, 1.70) | 0.751 |

*0h<Weekends CUS duration≤1h, 1h<Weekends CUS duration≤2h.

Model 1 was crude model.

Model 2 was adjusted age, sex, marital status, PIR, educational level, obesity.

Model 3 was adjusted age, sex, marital status, PIR, educational level, obesity, smoke, alcohol use, sleep trouble, PA, sleep duration in work day.

*indicates statistical significance.

In addition to phenotypic age, we also explored the relationships between CUS and several indicators previously associated with cognitive aging (fasting blood glucose, HbA1c, Homeostatic model assessment of insulin resistance, insulin levels, low-density lipoprotein cholesterol, and CRP). CUS was negatively correlated with fasting blood glucose, HbA1c, and, but showed no association with insulin levels, low-density lipoprotein cholesterol, or CRP (S3 Fig in S1 File).

**Bedtime and sleep duration in relation to CUS**

We first investigated the relationship between bedtime and aging (S1 Table in S1 File). Compared to participants who sleep earlier than 00:00, participants who sleep later face a significantly increased risk of aging, both on weekdays (OR=1.41, 95% CI: 1.11–1.79) and on weekends (OR=1.21, 95% CI: 1.01–1.45). This suggests that delayed

bedtime—regardless of whether it occurs on weekdays or weekends—is associated with a higher risk of aging, with a more pronounced effect observed on weekdays (41% increased risk) compared to weekends (21% increased risk).

After stratifying by bedtime (Table 3), individuals who sleep early on weekdays and engage in CUS on weekends had a decreasing trend in aging risk (OR=0.73, 95% CI: 0.52–1.02). However, when further subdividing by duration, only those who both sleep early and have a CUS duration of 0–1 hours (OR=0.68, 95% CI: 0.48–0.96) and 1–2 hours (OR=0.62, 95% CI: 0.48–0.81) exhibited a reduced risk of aging, not those who sleep late ($OR_{0-1h}$=0.95, 95%CI: 0.55–1.62, $OR_{1-2h}$=1.01, 95%CI: 0.54–1.92, $OR_{>2h}$=0.81, 95%CI: 0.55–1.20). The protective effect of CUS on aging risk is conditional on a regular early bedtime; CUS does not appear to mitigate aging risk in individuals with delayed bedtime habits. Additionally, we investigated differences in weekend bedtime (S2 Table in S1 File). Similar to previous findings, only participants who sleep early on weekends and have a CUS duration of 0–1 hour were associated with a reduced risk of aging (OR=0.65, 95% CI: 0.45–0.95), corresponding to a 35% lower risk. This reinforces that early bedtime—whether on weekdays or weekends—may be a prerequisite for CUS to exert beneficial effects on aging. Interaction analysis showed a significant interaction between CUS and weekday sleep duration (p for interaction=0.026). Furthermore, we combined bedtime and CUS duration to explore their relationship with aging (Fig 2). Compared to individuals who sleep early without CUS, only those who sleep early with a moderate amount of CUS (0–2 hours) showed a reduced risk of aging, while individuals with prolonged early sleep CUS (>2 hours) or late sleep did not exhibit a lower risk of aging.

Subsequently, we also observed the relationship between sleep duration and aging. RCS analysis revealed a U-shaped nonlinear relationship between sleep duration and the risk of aging (S4 Fig in S1 File). Compared to participants who sleep 7–8 hours, those who sleep more than 8 hours have a higher risk of aging (S3 Table in S1 File). The combined analysis shows that, compared to participants who sleep 7–8 hours and engage in catch-up sleep, those who sleep too little or too much without catch-up sleep have a significantly increased risk of aging (S4 Table in S1 File).

Compared to individuals without CUS, only those who usually sleep 7−8 hours exhibited a reduced risk of aging with CUS (OR=0.78, 95% CI: 0.58–0.96). However, for individuals with either excessively long sleep durations (OR=0.79, 95% CI: 0.48–1.30) or too short sleep durations (OR=1.23, 95% CI: 0.85–1.77), CUS did not improve their risk of aging. After further subdividing CUS duration, only individuals with usual sleep durations of 7−8 hours and CUS durations of 1−2 hours showed a reduced risk of aging (OR=0.47, 95% CI: 0.36–0.76) (Table 4). Mediation analysis showed that CUS mediated 14.5% of the relationship between sleep duration and aging, and 1.99% of the relationship between sleep timing and aging (S5 Fig in S1 File), so CUS may partially explain how sleep duration influences aging risk.

Table 3. The relationship between CUS and aging risk at different bedtime in workdays.

| | Before 0:00 at workdays | | After 0:00 at workdays | |
|---|---|---|---|---|
| **Weekend CUS** | | P | | P |
| No | Ref. | | Ref. | |
| Yes | 0.73 (0.52, 1.02) | 0.060 | 0.93 (0.57, 1.52) | 0.694 |
| Weekend CUS duration (h) | | | | |
| ≤0 | Ref. | | Ref. | |
| 0-1* | **0.68(0.48, 0.96)** | **0.036*** | 0.95(0.55, 1.62) | 0.805 |
| 1-2* | **0.62(0.48, 0.81)** | **0.004*** | 1.01(0.54, 1.92) | 0.958 |
| >2 | 0.95(0.49, 1.87) | 0.863 | 0.81(0.55, 1.20) | 0.236 |

*0h<Weekend CUS duration≤1h, 1h<Weekend CUS duration≤2h.

Model was adjusted age, sex, marital status, PIR, educational level, obesity, smoke, alcohol use, sleep trouble, PA, sleep duration in work day.

*Indicates statistical significance.

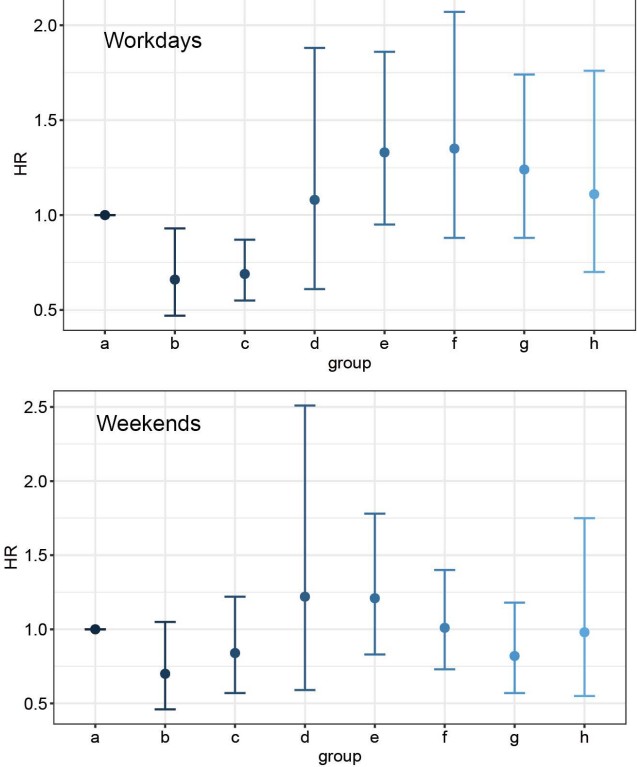

**Fig 2. Joint analysis of CUS and different bedtime regarding the aging risk.**

**Table 4. The relationship between CUS and aging risk at different sleep duration in workdays.**

| Sleep duration in workdays (h) | <7 | | 7-8 | | >8 | |
|---|---|---|---|---|---|---|
| **Weekend CUS** | | **P** | | **P** | | **P** |
| No | Ref. | | Ref. | | Ref. | |
| Yes | 0.81 (0.50, 1.32) | 0.349 | **0.75 (0.58, 0.96)** | **0.030\*** | 0.79 (0.48, 1.30) | 0.304 |
| Weekend CUS duration (h) | | | | | | |
| ≤0 | Ref. | | Ref. | | | |
| 0-1* | 0.52 (0.27,1) | 0.050 | 0.83 (0.63, 1.09) | 0.144 | 0.72 (0.38, 1.37) | 0.260 |
| 1-2* | 0.99 (0.65, 1.51) | 0.972 | **0.47 (0.36, 0.76)** | **0.009\*** | 0.81 (0.32, 2.06) | 0.606 |
| >2 | 0.73 (0.39, 1.38) | 0.272 | 1.09 (0.53, 2.24) | 0.778 | 1.20 (0.49, 2.95) | 0.643 |

*0h<Weekends CUS duration≤1h, 1h<Weekends CUS duration≤2h.

Model was adjusted age, sex, marital status, PIR, educational level, obesity, smoke, alcohol use, sleep trouble, PA, sleep duration in work day.

*Indicates statistical significance.

## Subgroup analysis and sensitivity analysis

We conducted subgroup analyses based on the presence of sleep disorders and retirement status (S5 Table in S1 File). The results indicate that among individuals without sleep disorders, CUS reduced the risk of aging (OR=0.75, 95% CI: 0.56−1), and a CUS duration of 1−2 hours (OR=0.69, 95% CI: 0.49–0.96) was associated with the optimal reduction in the

risk of aging. In individuals who were not retired, a CUS duration of 1−2 hours (OR=0.71, 95% CI: 0.51–0.98) was also effective in lowering their risk of aging.

Considering significant differences in compensatory sleep patterns across different age groups, we further supplemented an analysis of age subgroups (S6 Table in S1 File). The results showed that compensatory sleep was associated with a reduced risk of aging in the subgroups of age < 30 (OR=0.63, 95%CI: 0.54–0.74), 30 ≤ Age < 40 (OR=0.67, 95%CI: 0.53–0.84), and 40 ≤ Age < 50 (OR=0.78, 95%CI: 0.62–0.97).

Sensitivity analysis indicated that after excluding participants taking sleep and psychiatric medications, the results remained robust (S7 Table in S1 File). Similar to the main findings, CUS continued to reduce the risk of aging in these participants (OR=0.77, 95% CI: 0.59–0.98), with a 23% lower risk, especially in those with CUS durations of 0−1 hour (OR=0.73, 95% CI: 0.55–0.95; 27% lower risk) and 1−2 hours (OR=0.69, 95% CI: 0.54–0.88; 31% lower risk). This confirms that the association between CUS and reduced aging risk is not confounded by sleep or psychiatric medications. Considering that some participants provided work schedules, we adjusted their work schedules and re-analyzed, or excluded some participants who worked night shifts. The results still showed that getting CUS within 2 hours had an effect on reducing the risk of aging, further supporting the stability of our findings.

## Discussion

In this large cross-sectional study, we found that weekend CUS, especially when done in a reasonable manner (0–1 hour and 1–2 hours), can reduce the risk of aging in participants. However, excessively long CUS durations were not associated with improved aging. Further investigations revealed that CUS can improve the risk of aging in participants who usually have good sleep habits, including early bedtime (before 0:00) and normal sleep duration (7–8 hours). On the other hand, if participants usually experience sleep disorders, sleep late, or have abnormal sleep durations, weekend CUS was not associated with an improvement in their aging risk.

Although there is no direct evidence on the relationship between CUS and aging, some studies have investigated the relationship between sleep patterns and aging. A sleep index composed of sleep type, snoring, daytime sleepiness, sleep duration, insomnia, and difficulty waking up is negatively correlated with accelerated biological aging. This implies that the better the sleep quality, the lower the risk of aging for participants. The authors also found that good sleep quality can even offset the aging risk associated with air pollution [28]. Wang et al. also investigated the relationship between sleep parameters and biological age using the NHANES database. They found that greater sleep variability, more frequent irregular sleep patterns, and increased social jet lag were associated with an older biological age [29]. In addition, many studies have explored the relationship between Weekend CUS and components used to calculate biological age. Jang et al, after analyzing over 4,000 Koreans, reported that males with CUS of less than two hours on the weekends had a lower risk of abnormal blood lipid levels [30]. This is consistent with the findings of this study, suggesting that short periods of CUS may be more beneficial for the recovery of lipid metabolism, the authors also pointed out that normal sleep duration could reduce the risk of abnormal blood lipid levels. For another component, CRP, similar results were found. Han et al indicated a significant association between Weekend CUS and low-sensitivity CRP. Both longer and shorter durations of CUS were found to be unrelated to elevated CRP levels [31]. Another study specifically investigated the relationship between Weekend CUS and metabolic disorders, in addition to hypertension, Weekend CUS was found to be independently associated with obesity, type 2 diabetes, and high cholesterol levels [32]. In summary, these diseases are closely related to aging, and to some extent, they align with our study findings. However, some results differ from previous research. A study systematically investigated the relationship between CUS patterns and factors such as energy intake and insulin. They found that while there are benefits to Weekend CUS, such as restoring insulin sensitivity in the liver and muscles, this effect is short-lived, and irregular sleep following CUS can lead to delayed energy intake and weight gain [33]. The reasons for these differences may be attributed to variations in the definition of the reference group and differences in sample size and observation time in the study populations.

Interestingly, CUS exhibits the strongest protective effect against aging in participants who sleep 7–8 hours on workdays, suggesting that a regular lifestyle and periodic relief from work stress may contribute to a lower risk of aging. Sleep homeostasis may merely be a reflection of social stress [34]. Pan et al. demonstrated that stressful life events are significant factors leading to poor sleep quality, and interventions to reduce stress can effectively address sleep disorders [35]. This implies that maintaining a regular lifestyle and managing social stress may be the fundamental reasons for resisting aging.

From a mechanistic perspective, the relationship between irregular sleep, CUS, and aging is traceable. Sleep deprivation activates inflammatory signaling pathways, involving nuclear factor-κB, activating protein 1, and the signal transducer and activator of transcription family of proteins, which increases mRNA levels encoding pro-inflammatory cytokines [36,37]. Inflammation exacerbates cellular senescence, mitochondrial dysfunction, and DNA damage, consequently accelerating the aging process and increasing the phenotypic age [38]. On the other hand, circadian rhythm changes are also an important factor in explaining the relationship between CUS and aging. A regular circadian rhythm is a crucial condition for maintaining biological adaptation and adapting to the environment. Animal experiments indicate that mismatch between the internal biological clock and daily environmental changes is detrimental to survival [39]. The decline in aging and circadian rhythm function is mutually associated and mutually reinforcing. Normal aging is also accompanied by the decline in the function of the suprachiasmatic nucleus, DNA replication and hormonal changes, leading to disturbances in circadian rhythm [40,41]. This is because the peak of oxidation occurs during the day, followed by an increase in reactive oxygen species levels, while the peak of DNA replication happens at night. Normal circadian rhythm temporarily isolates these two processes to ensure that DNA replication occurs when ROS levels are at their lowest [42,43]. This also explains the reasons for the increased risk of aging in individuals with excessively long or short sleep durations in this study, and why the risk of aging does not continuously decrease in individuals with excessively long CUS durations.

Certainly, there are some limitations to our study. Firstly, it is a cross-sectional study, the uncertainty of causation is inevitable [44,45]. We cannot accurately determine whether CUS leads to aging or whether the biomarkers used in the definition of biological age affect sleep quality, forcing participants to get extra sleep. Secondly, participants' sleep information relies on self-reports rather than electronic device collection. However, self-reporting has the advantages of convenience and wide applicability, making it suitable for large-scale population studies like ours. NHANES lacks information on participants living with children or elderly people, and taking care of children or elderly people with limited mobility often leads to irregular sleep patterns [46,47]. Additionally, sleep disorders in the sensitivity analysis were only based on physician diagnosis, and no specific types of sleep disorders (such as OSA) were provided. Finally, the impact of some potential confounding factors, such as dietary patterns and exposure to toxins, should also be considered.

## Conclusion

This large cross-sectional study suggests that, compared to participants without weekend CUS, having 0–2 hours of CUS on weekends contributes to a reduced risk of aging. Interestingly, the anti-aging benefits of CUS are more pronounced in individuals who habitually go to bed early (before 0:00) and have a normal sleep duration (7–8 hours). However, for those with irregular sleep patterns during weekdays, the anti-aging effects of CUS appear to be minimal. These findings provide new insights into anti-aging strategies and sleep management.

## Supporting information

**S1 File. Supplementary method, figures and tables.**
(DOCX)

## Author contributions

**Conceptualization:** Nan Yao, Guoyong Yu.

**Data curation:** Nan Yao.

**Formal analysis:** Linrui Qi.

**Funding acquisition:** Linrui Qi.

**Investigation:** Chenan Liu, Jun Qu.

**Methodology:** Liuyi Shen, Chenan Liu, Ning Duan, Jun Qu.

**Project administration:** Ning Duan.

**Resources:** Liuyi Shen, Linrui Qi, Wenqiang Li, Fuzhou Han.

**Software:** Nan Yao, Liuyi Shen, Linrui Qi, Wenqiang Li, Fuzhou Han.

**Supervision:** Nan Yao, Guoyong Yu, Jun Qu.

**Validation:** Nan Yao, Liuyi Shen, Wenqiang Li, Guoyong Yu, Jun Qu.

**Visualization:** Nan Yao, Wenqiang Li, Chenan Liu, Fuzhou Han.

**Writing – original draft:** Nan Yao.

**Writing – review & editing:** Liuyi Shen, Linrui Qi, Fuzhou Han, Guoyong Yu, Jun Qu.

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
