## [Decision Letter · Decision Letter 0]

10 Mar 2025

Dear Dr. Qu,

Thank you for submitting your manuscript to PLOS ONE. After careful consideration, we feel that it has merit but does not fully meet PLOS ONE’s publication criteria as it currently stands. Therefore, we invite you to submit a revised version of the manuscript that addresses the points raised during the review process.

The study is interesting, well-articulated, and has the potential to contribute to sleep science. However, several areas require clarification. As a result, I consider the manuscript to have**‘Risky major revision’** due to its limitations. In addition to Reviewer 1’s comments, the author may also want to address the following points

**Abstract:** In addition to stating the OR, the author may consider providing an interpretation (e.g., 20% lower odds [OR 0.8, 95% CI: 0.63–1]). **Intro:** A more comprehensive literature review is recommended to better establish the rationale for the study, particularly regarding Weekend Catch-Up Sleep and the Risk of Aging. **Exposure Assessment:** I may have missed it, but I did not see the paper “[5] Kim et al. (2021),” which assessed the relationship between weekend CUS and other factors such as phenotypic age and anti-aging effects. Please clarify. **Statistical Analysis:** The authors may consider explaining the rationale for adjusting for age when age itself is the outcome variable. If aging (phenotypic age) is used as the outcome, why was self-reported age included as an adjustment variable? Similarly, for other adjusted variables, how do they justify their inclusion in the model? For example, if education and BMI were not significantly associated, what was the rationale for including them? **Results:** Providing an interpretation of the results would enhance the paper’s readability. **Conclusion:** Is this a cohort study? The author previously described it as cross-sectional—please clarify. **Tables: ** Authors may consider formatting the mean and SD as Age (mean±sd); Could you please consider indicating which tests were undertaken for which variables in the table, may be by using the endnote? I mean which mean diff was t-test and which one is Mann-Whitney U tests; Indicating the sig. using * would be useful.

We look forward to receiving your revised manuscript.

Kind regards,

Fakir Md Yunus, PhD, MSC, MPH, MBBS

Academic Editor

PLOS ONE

Journal Requirements:

2. In this instance it seems there may be acceptable restrictions in place that prevent the public sharing of your minimal data. However, in line with our goal of ensuring long-term data availability to all interested researchers, PLOS’ Data Policy states that authors cannot be the sole named individuals responsible for ensuring data access (http://journals.plos.org/plosone/s/data-availability#loc-acceptable-data-sharing-methods ).

Before we proceed with your manuscript, please also provide non-author contact information (phone/email/hyperlink) for a data access committee, ethics committee, or other institutional body to which data requests may be sent. If no institutional body is available to respond to requests for your minimal data, please consider if there any institutional representatives who did not collaborate in the study, and are not listed as authors on the manuscript, who would be able to hold the data and respond to external requests for data access? If so, please provide their contact information (i.e., email address). Please also provide details on how you will ensure persistent or long-term data storage and availability

Reviewers' comments:

Reviewer's Responses to Questions

**Comments to the Author**

1. Is the manuscript technically sound, and do the data support the conclusions?

Reviewer #1: Yes

Reviewer #2: Yes

2. Has the statistical analysis been performed appropriately and rigorously?

Reviewer #1: Yes

Reviewer #2: Yes

3. Have the authors made all data underlying the findings in their manuscript fully available?

Reviewer #1: Yes

Reviewer #2: Yes

4. Is the manuscript presented in an intelligible fashion and written in standard English?

Reviewer #1: Yes

Reviewer #2: Yes

Reviewer #1: In this study, the authors investigate the relationship between weekend Catch-Up Sleep (CUS) and biological aging using data from NHANES 2017-2018. The study presents an interesting perspective on how sleep compensation on weekends may influence aging risk, particularly in individuals with different sleep habits and durations. While the findings are intriguing, several methodological and conceptual concerns need to be addressed to strengthen the manuscript.

Major Concerns:

Major Comment #1: Specification Regarding Sleep Disorders, specially Sleep Apnea

The manuscript considers sleep disorders in subgroup and sensitivity analyses but does not specify which disorders are included. Given that NHANES 2017-2018 contains specific questions on obstructive sleep apnea (OSA), snoring, and breathing interruptions during sleep, it is unclear whether these factors were considered. OSA is particularly relevant as it is linked to inflammation, metabolic dysfunction, and accelerated biological aging—factors that could confound the observed relationship between CUS and aging.

Additionally, while the manuscript states that sleep disorder information was obtained from NHANES, it appears to rely on the broad self-reported question about "trouble sleeping." This approach does not distinguish between insomnia, OSA, or restless legs syndrome, which may have differing impacts on aging.

The authors should clarify whether:

- OSA was included as a distinct category in the analysis. If not, this is a significant limitation that should be acknowledged.

- Snoring and breathing irregularities were considered as proxies for undiagnosed OSA.

- Additional subgroup analyses could be performed, distinguishing between different sleep disorder types.

Furthermore, the sensitivity analysis excludes participants taking sleep and psychiatric medications, but the manuscript does not specify which medications were excluded. Since benzodiazepines, non-benzodiazepine hypnotics, and antidepressants can alter sleep architecture and affect biological aging, this exclusion criterion should be explicitly defined.

Major Comment #2: Mediation and Moderation Analyses

The current statistical approach categorizes weekend CUS duration into discrete groups (e.g., no CUS, 0-1h, 1-2h, >2h), which may limit the ability to detect more nuanced relationships. A more robust approach would be to apply moderation and mediation analyses, which could provide deeper insight into the role of CUS in aging.

- Mediation Analysis: CUS duration could act as a mediator between weekday sleep patterns (e.g., total weekday sleep, bedtime) and aging risk. This would help determine whether the protective effects of early bedtime or sufficient weekday sleep are explained through CUS.

- Moderation Analysis: Rather than treating CUS as a fixed categorical variable, it could be modeled as a moderator of the relationship between weekday sleep duration and aging. This would clarify whether CUS can buffer the negative effects of insufficient weekday sleep.

Applying these approaches would allow for a more continuous and dynamic understanding of sleep patterns rather than relying on static categorical cutoffs. If such analyses are not feasible with the current dataset, the authors should discuss this limitation in the manuscript.

Major Comment #3: Age-Stratified Analysis

The manuscript investigates the relationship between weekend CUS and aging but does not explicitly differentiate between young and older adults, either biologically or chronologically. While the authors use phenotypic age as a biomarker for biological aging, they do not analyze whether the effects of CUS differ across different age groups. This omission is important because aging is a continuous process, and the impact of sleep compensation on biological age might differ between younger and older individuals.

For example, the manuscript notes that CUS prevalence peaks in the 30-40 age group before declining, yet no further stratification is conducted to examine whether CUS has a differential effect on aging risk in younger versus older adults. A subgroup analysis based on age deciles or biologically defined aging categories (e.g., younger adults vs. middle-aged vs. older adults) would strengthen the findings and provide greater clarity on how CUS interacts with aging processes at different life stages.

Major Comment #4: Consideration for Cognitive Aging

The manuscript focuses on biological aging but does not address whether weekend CUS is related to cognitive aging. While NHANES 2017-2018 does not include direct cognitive tests it does provide metabolic and inflammatory biomarkers that could serve as indirect markers of cognitive decline.

Given the well-established link between sleep, aging, and cognitive function, the authors should discuss whether their aging measure captures cognitive aspects or is limited to metabolic/physiological aging. If possible, additional analyses could examine how CUS relates to metabolic and inflammatory markers that are relevant for cognitive aging.

Major Comment #5: Limited Citations

The introduction primarily relies on single references per claim, which may limit the depth of evidence supporting its conclusions. While citing individual studies is valuable, certain claims—especially those regarding the effects of sleep and aging—would benefit from referencing multiple sources to provide a broader scientific foundation.

To strengthen the introduction, the authors should increase citation density by incorporating multiple studies for each major claim.

Reviewer #2: The study is solid overall, the topic is relevant, and the work is well-executed. The NHANES 2017-2018 data provide a strong foundation, the statistical analysis is appropriately conducted, logistic regression is used correctly, and the conclusions align with the presented results. But there are a few things that could be clarified. First, at times, the authors seem to imply a causal relationship, even though the study is clearly correlational. It would be good to soften some of the wording because it’s still unclear whether catch-up sleep itself directly affects aging processes or if it simply correlates with other factors that do. Second, regarding data availability—everything is stated correctly, but it would be helpful to specify exactly where the raw data or more detailed summary statistics can be accessed, especially if any preprocessing was applied, since that always adds transparency. The finding that the effect is mainly seen in people who go to bed before midnight is particularly interesting, and it would be great if the authors could elaborate on possible explanations—are there physiological mechanisms, behavioral factors, or something else at play? The language of the manuscript is clear overall, but the structure of the results section could be slightly improved. Sometimes, transitions between statistical models feel abrupt, which might make it harder to follow for readers who are not fully immersed in the details. In general, this is a well-conducted and interesting study, but adding a bit more clarity on these points would make it even stronger.

**Do you want your identity to be public for this peer review?** For information about this choice, including consent withdrawal, please see our Privacy Policy

Reviewer #1: **Yes: ** Daniel Baena

Reviewer #2: **Yes: ** Denis Banchenko

---

## [Author Response · Author response to Decision Letter 1]

19 Jun 2025

Dear editors and reviewers,

Warm greetings of the day and hope this finds you well.

Thank you for your recognition and comments on our study. We have carefully discussed and reviewed each comment, making point-by-point revisions accordingly. Below is our response:

1.Response to editor:

Thank you for providing the following Data availability statement:

"Jun Qu had full access to all the data in the study and takes responsibility for the integrity of the data and the accuracy of the data analysis."

Before we can proceed, please respond to the following queries in the "Author Comments" box or in your Cover Letter:

1a) Please clarify whether your underlying data are restricted for ethical or legal reasons or if you are able to publicly share a de-identified data set.

1b) If the underlying data can be shared publicly and are NOT restricted, please upload your data set to a public repository and provide us with a URL/DOI or upload the data in your manuscript’s Supporting Information files. For a list of recommended repositories, some of which are able to hold sensitive data, please see here: https://journals.plos.org/plosone/s/recommended-repositories)

1c) If the data are restricted, please explain these restrictions in detail. For more information on what we consider acceptable restrictions to publicly sharing data, please see our guidelines: https://journals.plos.org/plosone/s/data-availability#loc-acceptable-data-access-restrictions.

1d) If the data is restricted for legal or ethical reasons, please provide contact information for a data access committee, ethics committee, or other institutional body to which data requests may be sent. If data are owned/restricted by a third party, please indicate how others may request data access.

Response: All data are derived from public databases. Therefore, I cannot upload public data as my own, but I can disclose how to apply for access. The revised content for the "Access to data and data analysis" section is as follows: All data can be applied for through the official website of NHANES (.https://wwwn.cdc.gov/nchs/nhanes/Default.aspx).

In this study, the authors investigate the relationship between weekend Catch-Up Sleep (CUS) and biological aging using data from NHANES 2017-2018. The study presents an interesting perspective on how sleep compensation on weekends may influence aging risk, particularly in individuals with different sleep habits and durations. While the findings are intriguing, several methodological and conceptual concerns need to be addressed to strengthen the manuscript.

Major Concerns:

Major Comment #1: Specification Regarding Sleep Disorders, specially Sleep Apnea

The manuscript considers sleep disorders in subgroup and sensitivity analyses but does not specify which disorders are included. Given that NHANES 2017-2018 contains specific questions on obstructive sleep apnea (OSA), snoring, and breathing interruptions during sleep, it is unclear whether these factors were considered. OSA is particularly relevant as it is linked to inflammation, metabolic dysfunction, and accelerated biological aging—factors that could confound the observed relationship between CUS and aging.

Additionally, while the manuscript states that sleep disorder information was obtained from NHANES, it appears to rely on the broad self-reported question about "trouble sleeping." This approach does not distinguish between insomnia, OSA, or restless legs syndrome, which may have differing impacts on aging.

The authors should clarify whether:

OSA was included as a distinct category in the analysis. If not, this is a significant limitation that should be acknowledged.

Snoring and breathing irregularities were considered as proxies for undiagnosed OSA.

Additional subgroup analyses could be performed, distinguishing between different sleep disorder types.

Response: Thank you for your comments.

Our diagnosis of sleep disorders was derived from the questionnaire question: "SLQ050 - Ever told doctor had trouble sleeping?" All sleep disorders were diagnosed by physicians, but specific types were not listed. This is part of the limitations of the NHANES database. However, as you mentioned, obstructive sleep apnea (OSA) is of particular concern because it is associated with inflammation, metabolic dysfunction, and accelerated biological aging. To address this issue, we made three compensations. First, we added a relevant description in the limitations section (Additionally, sleep disorders in the sensitivity analysis were only based on physician diagnosis, and no specific types of sleep disorders(such as OSA) were provided.). Second, we supplemented correlation analyses of indicators such as inflammation and metabolic dysfunction to clarify the mechanisms through which CUS affects aging (Figure S3). Third, we used frequent snoring (>2 times/week) as a surrogate marker for OSA and re - performed subgroup analyses to demonstrate the effects of different patterns of sleep disorders on the relationship between CUS and aging (Table S5).

Furthermore, the sensitivity analysis excludes participants taking sleep and psychiatric medications, but the manuscript does not specify which medications were excluded. Since benzodiazepines, non-benzodiazepine hypnotics, and antidepressants can alter sleep architecture and affect biological aging, this exclusion criterion should be explicitly defined.

Response: Thank you for your comments. In response to your comments, we have listed these drug types and supplemented some additional sensitivity analyses.

Original Revised

Sensitivity analysis involved excluding patients taking sleep medications and antipsychotic medications Sensitivity analysis involved excluding patients taking sleep medications and antipsychotic medications (psychotherapeutic agents, stimulants, glucocorticoid, benzodiazepines, non - benzodiazepine hypnotics, miscellaneous anxiolytics, sedatives and hypnotics, benzodiazepine anticonvulsants, phenothiazine antipsychotics,ssri antidepressants and hypnotics, antiemetics), participants with night shifts or participation without a work schedule provided, participants with CUS less than -30 minutes.

Major Comment #2: Mediation and Moderation Analyses

The current statistical approach categorizes weekend CUS duration into discrete groups (e.g., no CUS, 0-1h, 1-2h, >2h), which may limit the ability to detect more nuanced relationships. A more robust approach would be to apply moderation and mediation analyses, which could provide deeper insight into the role of CUS in aging.

Mediation Analysis: CUS duration could act as a mediator between weekday sleep patterns (e.g., total weekday sleep, bedtime) and aging risk. This would help determine whether the protective effects of early bedtime or sufficient weekday sleep are explained through CUS.

Moderation Analysis: Rather than treating CUS as a fixed categorical variable, it could be modeled as a moderator of the relationship between weekday sleep duration and aging. This would clarify whether CUS can buffer the negative effects of insufficient weekday sleep.

Applying these approaches would allow for a more continuous and dynamic understanding of sleep patterns rather than relying on static categorical cutoffs. If such analyses are not feasible with the current dataset, the authors should discuss this limitation in the manuscript.

Response: Thank you very much for your comments, which are extremely important. We conducted mediation and interaction analyses based on your suggestions. Surprisingly, a mediation effect does exist between them, and there is a significant interaction between CUS and sleep duration. We have included this content in both the supplementary materials and the main text. Thank you again for your comments!

Original Revised

None Methods: CUS duration could act as a mediator between weekday sleep patterns (e.g., total weekday sleep, bedtime) and aging risk. Therefore, mediational analysis was used to clarify the role of CUS in these relationships.

Interaction analysis was used to explore whether CUS is a moderator of the relationship between weekday sleep duration or weekday bedtime and aging.

Results: Interaction analysis showed a significant interaction between CUS and weekday sleep duration (p for interaction=0.026).

Mediation analysis showed that CUS mediated 14.5% of the relationship between sleep duration and aging, and 1.99% of the relationship between sleep timing and aging (Figure S5).

Major Comment #3: Age-Stratified Analysis

The manuscript investigates the relationship between weekend CUS and aging but does not explicitly differentiate between young and older adults, either biologically or chronologically. While the authors use phenotypic age as a biomarker for biological aging, they do not analyze whether the effects of CUS differ across different age groups. This omission is important because aging is a continuous process, and the impact of sleep compensation on biological age might differ between younger and older individuals.

For example, the manuscript notes that CUS prevalence peaks in the 30-40 age group before declining, yet no further stratification is conducted to examine whether CUS has a differential effect on aging risk in younger versus older adults. A subgroup analysis based on age deciles or biologically defined aging categories (e.g., younger adults vs. middle-aged vs. older adults) would strengthen the findings and provide greater clarity on how CUS interacts with aging processes at different life stages.

Response: Thank you for your comments. As you mentioned, there are significant differences in the patterns and levels of compensatory sleep across different age groups. In response to your comment, we supplemented the analysis of the association between CUS and aging in different age subgroups: <30, 30-40, 40-50, 50-60, and >60.The results showed that compensatory sleep was associated with a reduced risk of aging in the subgroups of age < 30 (OR=0.63, 95%CI: 0.54-0.74), 30≤Age<40 (OR=0.67, 95%CI: 0.53-0.84), and 40≤Age<50 (OR=0.78, 95%CI: 0.62-0.97).

Age<30 OR (95% CI) P

Weekends CUS

No Ref.

Yes 0.63 (0.54, 0.74) <0.001

30≤Age<40

Weekends CUS

No Ref.

Yes 0.67 (0.53, 0.84) 0.001

40≤Age<50

Weekends CUS

No Ref.

Yes 0.78 (0.62, 0.97) 0.030

50≤Age<60

Weekends CUS

No Ref.

Yes 0.84 (0.63, 1.12) 0.239

Age>60

Weekends CUS

No Ref.

Yes 0.81 (0.66, 1) 0.051

Major Comment #4: Consideration for Cognitive Aging

The manuscript focuses on biological aging but does not address whether weekend CUS is related to cognitive aging. While NHANES 2017-2018 does not include direct cognitive tests it does provide metabolic and inflammatory biomarkers that could serve as indirect markers of cognitive decline.

Given the well-established link between sleep, aging, and cognitive function, the authors should discuss whether their aging measure captures cognitive aspects or is limited to metabolic/physiological aging. If possible, additional analyses could examine how CUS relates to metabolic and inflammatory markers that are relevant for cognitive aging.

Response: Thank you for your comments. This is an interesting idea. We included indicators such as HbA1c, HOMA-IR, LDL-C, and CRP, and explored the relationships between CUS and these indicators from the aspects of glucose metabolism, insulin resistance, lipid metabolism, and inflammation, as these indicators have been proven to be associated with aging or cognitive decline. The results showed that the duration of CUS was associated with fasting blood glucose, HbA1C, and HOMA-IR, but not with lipid metabolism or inflammation indicators. This is interesting and seems to suggest that compensatory sleep may affect cognitive aging-related mechanisms by influencing the glucose metabolism of participants. We have supplemented this result in the supplementary material.

Major Comment #5: Limited Citations

The introduction primarily relies on single references per claim, which may limit the depth of evidence supporting its conclusions. While citing individual studies is valuable, certain claims—especially those regarding the effects of sleep and aging—would benefit from referencing multiple sources to provide a broader scientific foundation.

To strengthen the introduction, the authors should increase citation density by incorporating multiple studies for each major claim.

Response: Thank you for your comments. We have added some references in the introduction section, especially those related to the effects of sleep and aging, to make the entire argument more evidence-based. Details are as follows:

Original Revised

1.which includes the impact on cardiovascular diseases, cognition, metabolism, and various other conditions such as tumors[1].

2.Some prospective studies also indicated a close association between excessively long or short sleep duration and an increased risk of cancer[4].

3.Liu et al. demonstrated that individuals with 1 to 2 hours of CUS had a lower risk of depression compared to those who did not have CUS [9].

4.such as increased susceptibility to diseases, genomic instability, epigenetic changes, and chronic inflammation [10].

5.Up to this point, there hasn't been any research exploring the relationship between the increasingly prevalent habit of weekend CUS and aging. Therefore, this study aims to investigate the association between CUS and aging using the NHANES database. The research also explores the anti-aging effects of weekend CUS in patients with different sleep times, sleep durations, and the sleep disorders. 1.which includes the impact on cardiovascular diseases, cognition, metabolism, and various other conditions such as tumors[1,41].

2.Some prospective studies also indicated a close association between excessively long or short sleep duration and an increased risk of cancer[4,42].

3.Liu et al. demonstrated that individuals with 1 to 2 hours of CUS had a lower risk of depression compared to those who did not have CUS [9]. Chen et al. showed that compared with those without CUS, 2-3 hours of CUS was closely associated with a lower prevalence of chronic kidney disease[44].

4.such as increased susceptibility to diseases, genomic instability, epigenetic changes, and chronic inflammation [10,45].

5.Up to now, although no studies have explored the relationship between the increasingly common weekend catch-up sleep (CUS) habit and aging, research has investigated the association between sleep duration patterns and aging. Results show that compared with participants in the normal stable trajectory, trajectories of increased and short-term stable sleep duration are associated with a lower likelihood of successful aging, yet the relationship between CUS and aging remains unaddressed[46]. To fill this gap, this study aims to investigate the association between CUS and aging using the NHANES database, and further explore the anti-aging effects of weekend CUS among individuals with different sleep times, sleep durations, and sleep disorders.

The above are all our responses.

All the best.

Nan Yao.

---

## [Decision Letter · Decision Letter 1]

4 Aug 2025

Dear Dr. Qu,

Thank you for submitting your manuscript to PLOS ONE. After careful consideration, we feel that it has merit but does not fully meet PLOS ONE’s publication criteria as it currently stands. Therefore, we invite you to submit a revised version of the manuscript that addresses the points raised during the review process.

We look forward to receiving your revised manuscript.

Kind regards,

Fakir Md Yunus, PhD, MSC, MPH, MBBS

Academic Editor

PLOS ONE

Journal Requirements:

Additional Editor Comments:

Thank you for your thoughtful and thorough responses to the reviewers’ comments. The manuscript has significantly improved and is close to being ready for publication. To further strengthen the clarity, coherence, and overall readability of the paper, we encourage the authors to carefully review and consider the following editorial suggestions. These are intended to enhance the presentation and accessibility of your work for a broader readership. We look forward to receiving the revised version.

Abstract: In addition to stating the OR, the author may consider providing an interpretation (e.g., 20% lower odds [OR 0.8, 95% CI: 0.63–1]).

Intro: A more comprehensive literature review is recommended to better establish the rationale for the study, particularly regarding Weekend Catch-Up Sleep and the Risk of Aging.

Exposure Assessment: I may have missed it, but I did not see the paper “[5] Kim et al. (2021),” which assessed the relationship between weekend CUS and other factors such as phenotypic age and anti-aging effects. Please clarify.

Statistical Analysis: The authors may consider explaining the rationale for adjusting for age when age itself is the outcome variable. If aging (phenotypic age) is used as the outcome, why was self-reported age included as an adjustment variable? Similarly, for other adjusted variables, how do they justify their inclusion in the model? For example, if education and BMI were not significantly associated, what was the rationale for including them?

Results: Providing an interpretation of the results would enhance the paper’s readability.

Conclusion: Is this a cohort study? The author previously described it as cross-sectional—please clarify.

Tables: Authors may consider formatting the mean and SD as Age (mean±sd); Could you please consider indicating which tests were undertaken for which variables in the table, may be by using the endnote? I mean which mean diff was t-test and which one is Mann-Whitney U tests; Indicating the sig. using '*' would be useful.

Reviewers' comments:

Reviewer's Responses to Questions

**Comments to the Author**

Reviewer #1: All comments have been addressed

2. Is the manuscript technically sound, and do the data support the conclusions?

Reviewer #1: Yes

3. Has the statistical analysis been performed appropriately and rigorously?

Reviewer #1: Yes

4. Have the authors made all data underlying the findings in their manuscript fully available?

Reviewer #1: Yes

5. Is the manuscript presented in an intelligible fashion and written in standard English?

Reviewer #1: Yes

Reviewer #1: All my comments have been addressed. I have no further questions.

Great work!

**Do you want your identity to be public for this peer review?** For information about this choice, including consent withdrawal, please see our Privacy Policy

Reviewer #1: **Yes: ** Daniel Baena

---

## [Author Response · Author response to Decision Letter 2]

12 Aug 2025

Dear editor,

Thank you for your comments and recognition. We have noted that the reviewers have no further comments. Below is a point-by-point response to the editor's comments. We hope that all comments and issues can be addressed, making this manuscript more complete.

Additional Editor Comments:

Thank you for your thoughtful and thorough responses to the reviewers’ comments. The manuscript has significantly improved and is close to being ready for publication. To further strengthen the clarity, coherence, and overall readability of the paper, we encourage the authors to carefully review and consider the following editorial suggestions. These are intended to enhance the presentation and accessibility of your work for a broader readership. We look forward to receiving the revised version.

Abstract: In addition to stating the OR, the author may consider providing an interpretation (e.g., 20% lower odds [OR 0.8, 95% CI: 0.63–1]).

Response: Thank you for your comments; we have made corresponding revisions.

Intro: A more comprehensive literature review is recommended to better establish the rationale for the study, particularly regarding Weekend Catch-Up Sleep and the Risk of Aging.

Response: Thank you for your comments. There is indeed a scarcity of literature reviews on catch-up sleep and aging. We have replaced it with a more comprehensive literature review related to aging and catch-up sleep.

Exposure Assessment: I may have missed it, but I did not see the paper “[5] Kim et al. (2021),” which assessed the relationship between weekend CUS and other factors such as phenotypic age and anti-aging effects. Please clarify.

Response: Thank you for this comment. We sincerely apologize for the incorrect order of the literature citations due to issues with the EndNote software. This has now been corrected.

Statistical Analysis: The authors may consider explaining the rationale for adjusting for age when age itself is the outcome variable. If aging (phenotypic age) is used as the outcome, why was self-reported age included as an adjustment variable? Similarly, for other adjusted variables, how do they justify their inclusion in the model? For example, if education and BMI were not significantly associated, what was the rationale for including them?

Response: Thank you for this insightful comment. Regarding the inclusion of self-reported chronological age as an adjustment variable when phenotypic age (a measure of biological aging) is the outcome: Chronological age and phenotypic age, while correlated, represent distinct constructs—chronological age reflects the passage of time, whereas phenotypic age captures biological aging processes that may diverge from chronological age (e.g., individuals with the same chronological age can exhibit different rates of biological aging). By adjusting for chronological age, we aim to isolate the association between CUS and accelerated biological aging (i.e., phenotypic age relative to chronological age), rather than conflating it with the general aging process tied to time. This approach aligns with methodological precedents in aging research, where adjusting for chronological age helps disentangle biological aging from chronological progression.

For other adjusted variables (e.g., education, BMI), their inclusion was guided by theoretical and empirical evidence from prior literature, rather than solely by statistical significance in our dataset. Education is a well-established social determinant of health behaviors and aging trajectories, as it may influence health literacy, access to healthcare, and lifestyle factors (e.g., sleep patterns). BMI, as a marker of adiposity, has been consistently linked to biological aging processes, including inflammation and metabolic dysfunction, which are hypothesized to mediate aging risk. Even if these variables were not statistically significant in our analyses, their inclusion helps control for potential confounding—omitting them could introduce bias if they are associated with both CUS (exposure) and phenotypic age (outcome). This approach adheres to the principle of adjusting for a priori identified confounders to enhance the internal validity of our findings. We have supplemented the corresponding rationale in Supplementary Method 2 of the supplementary materials.

Results: Providing an interpretation of the results would enhance the paper’s readability.

Response: Thank you for your comment. We have added some interpretations and descriptions of the results part.

Conclusion: Is this a cohort study? The author previously described it as cross-sectional—please clarify.

Response: Thank you for your comments; we have made corresponding revisions.

Tables: Authors may consider formatting the mean and SD as Age (mean±sd); Could you please consider indicating which tests were undertaken for which variables in the table, may be by using the endnote? I mean which mean diff was t-test and which one is Mann-Whitney U tests; Indicating the sig. using '*' would be useful.

Response: Thank you for your comments; we have made corresponding revisions.

These are all our responses, and we hope they can address the above comments.

All the best.

Nan Yao.

---

## [Editor Report · Decision Letter 2]

3 Sep 2025

Relationship between Weekends Catch-Up Sleep and Risk of Aging

PONE-D-24-50221R2

Dear Dr. Qu,

We’re pleased to inform you that your manuscript has been judged scientifically suitable for publication and will be formally accepted for publication once it meets all outstanding technical requirements.

Kind regards,

Fakir Md Yunus, PhD, MSC, MPH, MBBS

Academic Editor

PLOS ONE
---

## [Editor Report · Acceptance letter]

PONE-D-24-50221R2

PLOS ONE

Dear Dr. Qu,

I'm pleased to inform you that your manuscript has been deemed suitable for publication in PLOS ONE. Congratulations! Your manuscript is now being handed over to our production team.

Kind regards,

on behalf of

Dr. Fakir Md Yunus

Academic Editor

PLOS ONE